# Intelligent Recognition of Chirp Radar Deceptive Jamming Based on Multi-Pulse Information Fusion

**DOI:** 10.3390/s21082693

**Published:** 2021-04-11

**Authors:** Xuegang Lan, Tao Wan, Kaili Jiang, Ying Xiong, Bin Tang

**Affiliations:** School of Information and Communication Engineering, UEST of China, Chengdu 611731, China; taowan.uestc0939@foxmail.com (T.W.); jiangkelly@foxmail.com (K.J.); xiongy@uestc.edu.cn (Y.X.); bint@uestc.edu.cn (B.T.)

**Keywords:** chirp radar, deceptive jamming, jamming identification, CNN, multi-pulse information fusion

## Abstract

The perception of jamming types is very important for protecting our radar in complex electromagnetic environments. Radar active deceptive jamming based on digital radio frequency memory (DRFM) has a high coherence with the target echo, which confuses the information of the target echo and achieves the effect of hiding the real target. Traditional deceptive jamming recognition methods need to extract complex features and artificially set classification thresholds, which is inefficient. The existing neural network-based jamming identification methods still follow the pattern of signal modulation-type identification, so there are fewer types of jamming that can be identified, and the identification accuracy is low in the case of low jamming-to-noise ratios (JNR). This paper studies the input of jamming recognition networks and proposes an improved intelligent identification method for chirp radar deceptive jamming. This method fuses three short-time Fourier transform time–frequency graphs disturbed by three consecutive pulse periods into a new graph as the input of the convolutional neural network (CNN). Using a CNN to classify the time–frequency image has realized the recognition of a variety of common deceptive jamming techniques. Similarly, by changing the network input, the original signal is used to replace the echo signal, which improves the accuracy of the jamming recognition in the case of a low JNR.

## 1. Introduction

With the development of technologies such as very large-scale integrated circuits (VLSI) and solid-state circuits, active deceptive jamming exhibits features such as high fidelity and high intelligence, which makes the electromagnetic environment of radar work worse and poses a huge threat to the survival of radar. In this context, radar systems have increasingly demanded active jamming identification methods.

In [1,2], the researchers studied the micro-Doppler features of real targets, and then extracted the time-varying Doppler as a feature parameter to distinguish deceptive jamming from the target echo. The researchers used the difference between the Doppler information obtained by the multi-spectrum measurement and the Doppler information obtained after the ranging calculation to identify range gate pull-off jamming and velocity gate pull-off jamming [3]. In [4], the researchers used the second- and fourth-order cumulants of the received signal of the radar as characteristic parameters to identify active jamming. In [5], the researchers studied and analyzed the influence of phase quantization on the digital radio frequency memory of the jamming, found that false level signals appeared in the towing jamming spectrum generated by a DRFM jammer, and using these factors as characteristics of the jamming, could further identify towing jamming.

Traditional jamming identification methods mostly follow the steps of jamming detection, jamming feature parameter extraction, and jamming pattern recognition. The related algorithms are greatly affected by the JNR and have poor robustness when facing target echo aliasing jamming, especially deceptive jamming with high coherence with the target echo. It is difficult to extract stable feature parameters with a high degree of discrimination from only the time and frequency domains, which will inevitably reduce the accuracy of the perception of jamming. Most traditional classification methods are non-adaptive classification methods such as statistical binary tree classification. The threshold setting of its key nodes is manually determined, and because the radar operator is often required to set it according to the radar display screen or judge based on their experience, the efficiency of the entire jamming identification process is low.

With the deepening of artificial intelligence technology research, deep learning networks can effectively uncover deep-level signal information, analyze conventional jamming features, extracting subtle features of jamming that are difficult to quantify, and intelligently identify jamming types. In 1998, Yann LeCun [6] and his colleagues proposed a convolutional neural network (CNN) model which had good performance in the field of handwritten digit recognition. A small number of researchers have made some attempts to apply CNN to the field of radar. In [7], the researchers used a combined network including CNN and support vector machine (SVM) networks to realize the recognition of the signal modulation type of low probability of intercept (LPI) radar and achieved good results. In [8,9,10,11], the researchers have all tried to use CNN network for radar jamming recognition. However, the above research was merely a simple application of simple neural networks. Radar jamming recognition has not been researched in depth, and the particularity of jamming recognition has not been considered. The radar signal modulation-type identification method is still used for jamming identification, and the input of the network is only a single-pulse jamming signal or a single-pulse jamming transform domain feature. Radar signal modulation-type identification can be performed from the intrapulse parameters of a single-pulse radar signal. However, the identification of deceptive jamming requires not only the intra-pulse characteristics of the jamming signal, but also the time difference and frequency difference between the jamming and the real signal. In addition to drag-type jamming, inter-pulse information needs to be interfered with to be identified.

In this article, we study the intelligent identification method of chirp radar active deceptive jamming. Through research on and the analysis of the common active deceptive jamming of radar, a jamming recognition model based on a convolutional neural network is proposed. Through the analysis of the jamming recognition model of the convolutional neural network, it is known that whether a neural network can correctly classify and recognize the jamming of the input depends on the amount of information contained in the inputs of the training and test sets. The existing neural network jamming recognition method has less input information and can only identify a few specific deceptive jamming techniques. Therefore, this paper studies the input of the neural network. By changing the input of the jamming identification network, the neural network recognizes a variety of common radar deceptive jamming techniques, and gains improved recognition accuracy. A short-time Fourier transform time–frequency diagram of the jamming is used as the input of the network to provide the time–frequency distribution information of the jamming. The echo signal is added to the network input so that the input includes the carrier frequency, bandwidth, modulation slope, and other true signal information required for jamming identification. The time–frequency diagram of multi-pulse jamming plus echo is used as the input of the recognition network, so that the input of the recognition network includes the interpulse information of the deceptive jamming. The recognition network is thus able to identify common chirp radar active deceptive jamming techniques, including repeating jamming and towing jamming. Using the original signal instead of the echo signal as the input of the network improves the success rate of the jamming recognition in the case of a low jamming-to-noise ratio (JNR).

## 2. Jamming Recognition Method

From the characteristics of radar active deceptive jamming, it can be seen that the way and mechanism of each kind of jamming are different, and their characteristic parameters in the time and frequency domains reflect the mathematical laws of the different dimensions. For linear frequency modulation (LFM) signal, because of its special time–frequency linear relationship, the time–frequency analysis method can more clearly reflect the characteristics of repeater jamming. Short-time Fourier transform (STFT) divides the signal into several time periods by windowing the signal in the time domain. The signal in each time period is regarded as a short-time stationary signal, the Fourier transform method is used to analyze it, and the frequency distribution of the corresponding time period is obtained. Compared to the traditional time-domain and frequency-domain characteristic parameters, the STFT spectrum of radar active deceptive jamming can directly reflect the jamming characteristics, as there is a very clear distinction between various kinds of jamming, which can be used for the subsequent identification of radar active deceptive jamming.

### 2.1. Common Chirp Radar Active Deceptive Jamming

Radar active deceptive jamming generates false targets by modulating and forwarding the transmitted radar signal so that the real target is hidden in a false target, and the information of the false target is used to disrupt the detection, parameter measurement, tracking, and positioning of the real target. Compared with suppression jamming, the main features of deceptive jamming are that it can obtain a larger signal processing gain and has strong coherence with the target echo in the time, frequency, and space domains. Common radar active deceptive jamming techniques include range gate pull-off jamming, velocity gate pull-off jamming, range- velocity gate pull-off jamming, velocity false target jamming, range false target jamming, intermittent sampling jamming, and smeared spectrum jamming. The expression of the LFM radar-transmitting signal is
(1)$$S\left(t\right)=rect(\frac{t}{T_{S}})\exp\left[j2\pi \left(f_{0}t+\frac{1}{2}\mu t^{2}\right)\right],$$
where *T_S_* is the time domain pulse width, *f*_0_ is the center frequency, *µ* = 2*π**B*/*T_S_* is the chirp constant, and B is the LFM bandwidth.

The false distance function generated by range gate pull-off (RGPO) [12] jamming is
(2)$$R_{j}\left(t\right)=\left\{\begin{array}{c} \begin{array}{cc} R, & 0\leq t<t_{1} \end{array}\\ \begin{array}{cc} R+v_{j}\left(t-t_{1}\right), & t_{1}\leq t<t_{2} \end{array}\\ \begin{array}{cc} close, & t_{2}\leq t<T_{j} \end{array} \end{array}\right\},$$
where R is the distance between the real target and the receiver, $v_{j}$ is the speed when towing at a constant speed, $t_{1}$ is the time when the towing starts, and $t_{2}$ is the time when the towing ends. Then, the forwarding delay of distance towing jamming relative to the true target echo is
(3)$$\Delta t_{j}\left(t\right)=\left\{\begin{array}{c} \begin{array}{cc} 0, & 0\leq t<t_{1} \end{array}\\ \begin{array}{cc} \frac{2v_{j}}{c}\left(t-t_{1}\right), & t_{1}\leq t<t_{2} \end{array}\\ \begin{array}{cc} close, & t_{2}\leq t<T_{j} \end{array} \end{array}\right\}.$$

In order to simplify the formula’s variables, without considering the jammer’s forwarding delay, the signal model of distance towing jamming is obtained as
(4)$$\begin{array}{ll} J_{RGPO}= & A_{j}\cdot rect\left(\frac{t-\Delta t_{j}\left(t\right)}{T_{S}}\right)\\ & \exp\left\{j2\pi \left[f_{0}\left(t-\Delta t_{j}\left(t\right)\right)+\frac{1}{2}\mu {\left(t-\Delta t_{j}\left(t\right)\right)}^{2}\right]\right\}, \end{array}$$
where $A_{j}$ is the amplitude of the jamming. Generally, the energy of the jamming is greater than the energy of the target echo.

When the towing speed of the distance gate towing jamming, $v_{j}$, is 0, distance towing jamming becomes range false target jamming (RFTJ) [13].

The Doppler frequency shift difference between velocity gate pull-off (VGPO) [14] jamming and the real target echo signal changes as
(5)$$f_{dj}=\left\{\begin{array}{c} \begin{array}{cc} f_{d}, & 0\leq t<t_{1} \end{array}\\ \begin{array}{cc} f_{d}+v_{f}\left(t-t_{1}\right), & t_{1}\leq t<t_{2} \end{array}\\ \begin{array}{cc} close, & t_{2}\leq t<T \end{array} \end{array}\right.,$$
where $f_{d}=2v_{r}/\lambda$, $v_{r}$ is the radial velocity of the target relative to the receiver, and $v_{f}$ (Hz/s) is the towing velocity. The signal model of speed towing jamming is shown in as
(6)$$J_{VGPO}=A_{j}\cdot rect\left(\frac{t}{T_{S}}\right)\exp\left\{j2\pi \left[\left(f_{0}+f_{dj}\right)t+\frac{1}{2}\mu t^{2}\right]\right\},$$
where $A_{j}$ is the magnitude of jamming.

When the towing velocity of velocity gate pull-off jamming, $v_{r}$, is 0, velocity gate pull-off jamming becomes velocity false target jamming (VFTJ) [15].

Range-velocity gate pull-off (R-VGPO) [16] jamming combines the distance gate and velocity gate pulls, and the signal model is established as
(7)$$\begin{array}{l} J_{RVGPO}=A_{j}\cdot rect\left(\frac{t-\Delta t_{j}\left(t\right)}{T_{S}}\right)\\ \;\;\exp\left\{j2\pi \left[\left(f_{0}+f_{dj}\right)\left(t-\Delta t_{j}\left(t\right)\right)+\frac{1}{2}\mu {\left(t-\Delta t_{j}\left(t\right)\right)}^{2}\right]\right\}. \end{array}$$

Interrupted-sampling directly jamming (ISDJ) [17] is generated by the DRFM jammer intercepting and forwarding the radar-transmitted signal and repeating the interception-forwarding process immediately after the forwarding is completed. The ISDJ generation process can be regarded as the process of sampling the radar signal at equal intervals and then forwarding it. It has the characteristics of a small jamming delay, and can obtain the pulse pressure gain of the original radar signal.

According to the working principle of ISDJ, above, its time domain expression is
(8)$$J_{\mathrm{ISDJ}}\left(t\right)=\sum_{n=0}^{N_{J}-1}rect\left(\frac{t-\left(2n+1\right)T_{W}}{T_{W}}\right)S\left(t-T_{W}\right),$$
where $S\left(t\right)$ is the radar signal intercepted by DRFM, $N_{J}$ is the number of slices, and $T_{W}$ is the slice width.

The process by which a DRFM jammer generates interrupted-sampling repeater jamming (ISRJ) [18] is similar to that of ISDJ, but the interception and forwarding strategies are different. After the DRFM jammer intercepts a segment of the radar signal, it delays and forwards the segment multiple times, and then repeats the interception delay forwarding process until the radar signal ends. The time domain expression of ISRJ is
(9)$$J_{\mathrm{ISRJ}}\left(t\right)=\sum_{m=1}^{M}\sum_{n=0}^{N_{J}-1}rect\left(\frac{t-\left(nM+n+m\right)T_{W}}{T_{W}}\right)S\left(t-mT_{W}\right),$$
where $S\left(t\right)$ is the radar signal intercepted by DRFM, $N_{J}$ is the number of slices, $T_{W}$ is the slice width, and M is the number of retransmissions.

Partial-pulse dense transmit jamming (PDTJ) [19] is generated by a DRFM jammer intercepting part of the radar signal, and then continuously forwarding this pulse to form a large number of dense false targets. Since the formation of PDTJ does not require a DRFM jammer to intercept and store a complete radar signal, it is easy to implement and has strong real-time performance. Its expression can be written as
(10)$$J_{\mathrm{PDTJ}}\left(t\right)=\sum_{m=1}^{M}\mathrm{rect}\left(\frac{t-mT_{W}}{T_{W}}\right)S\left(t-mT_{W}\right),$$
where $S\left(t\right)$ is the radar signal intercepted by DRFM, $T_{W}$ is the slice width, and M is the number of retransmissions.

Smeared spectrum (SMSP) jamming is a new distance false target jamming invented by Sparrow et al. in 2006 [20]. DRFM intercepts the chirp signal transmitted by the radar and stores it. After sampling it in the time domain, it uses a multiple of the original clock, frequency-extracts the sampled data, and then duplicates it to get SMSP jamming. According to the previous explanation, SMSP jamming is generated by using high-frequency sampling on the LFM signal transmitted by the radar, and the first sub-pulse can be expressed as
(11)$$J_{1}\left(t\right)=\exp\left(j\pi k^{\prime }t^{2}\right),k^{\prime }=nk,0\leq t\leq T_{s}/n,$$
where $k^{\prime }$ is the FM slope of the SMSP jamming. Copy $J_{1}\left(t\right)$ times to get the expression of SMSP jamming as
(12)$$J_{SMSP}\left(t\right)=\sum_{i=0}^{n-1}J_{1}\left(t-i\frac{T}{n}\right).$$

It can be seen from Equation (12) that a DRFM jammer compresses and replicates the intercepted radar transmission signal to obtain SMSP jamming. The spectrum of SMSP jamming is
(13)$$J_{SMSP}\left(f\right)=\frac{\mathrm{sinc}\left(fT\right)}{\sqrt{n}\mathrm{sinc}\left(fT/n\right)}\exp\left(-j\pi \left(1-\frac{1}{n}\right)T\right)S\left(\frac{f}{\sqrt{n}}\right),$$
where S(f) is the frequency spectrum of the radar-transmitted signal.

The expression of whole-pulse dense transmit jamming (WDTJ) [19] generated by the DRFM jammer is found in Equation (14). The jammer copies and stores a complete radar transmission signal, and then forwards it in a certain period.
(14)$$J_{WDTJ}\left(t\right)=S\left(t\right)⊗\left[\sum_{k=1}^{N}a_{k}\delta \left(t-\left(k-1\right)T_{j}\right)\right],$$
where $S\left(t\right)$ is the radar transmission signal, ⊗ is the convolution operation, $a_{k}$ is the modulation amplitude, usually with the constant value, $\delta \left(t\right)$, representing the impulse function, and $T_{j}$ is the jamming forwarding period.

### 2.2. Identification Framework and Network Model

CNN is a multi-level network model. In the field of image processing, its input is image data composed of multiple channels. After the data is subjected to multiple convolutions, pooling, and activation in the network, the features are extracted out, and then output through the fully-connected layer. The basic structure of CNN is shown in the Figure 1.

The training prediction process of the jamming recognition model is shown in Figure 2.

The jamming signal received by the radar was processed through a series of processing to obtain the input data of the neural network, and the data were divided into a training set, a verification set, and a test set. The training set was input into the training network with the CNN network as its core to obtain the network weights by iterative training. The verification set was used to verify the correct rate of network recognition and find the optimal network weight with the highest test accuracy. In the recognition process, the test set was input into the network model loaded with the optimal network weights for recognition.

In order to achieve higher recognition accuracy, this paper uses the ResNet50 [21] network in the deep residual network (ResNet) of the CNN as the core of the recognition model. ResNet was proposed by Microsoft Labs in 2015. With the help of the residual module in the network, it achieved an ultra-deep network structure (more than 1000 layers), and used Batch normalization to accelerate the training. ResNet won first place in the classification task and first place in target detection in the ImageNet competition that year. It also won first place in target detection and image segmentation in the COCO (Microsoft Common Objects in Context) dataset. COCO is a large and rich object detection, segmentation, and subtitle data set. It originated from the Microsoft COCO data set, which was founded by Microsoft in 2014. Like the ImageNet competition, it is regarded as one of the most significant competitions in the field of computer vision.

The parameters of the ResNet50 model are shown in Table 1. 

The ResNet50 model is shown in Figure 3. 

### 2.3. Network Input

The research of neural network and neural network training has been very mature, but the results obtained under different training and test inputs can very different even when processed by the same network. CNNs are widely used in the field of image recognition and classification. The use of a CNN for image classification requires distinct separable features between classes. Recognition based only on the time or frequency domain of deceptive jamming often fails to obtain correct results. Especially for LFM radar, deceptive jamming often has a special time–frequency linear relationship, and joint time–frequency characteristics can clearly reflect the characteristics of deceptive jamming. In this paper, short-time Fourier transform [22] results, which are widely used in signal time–frequency analysis, are used as the input for the recognition model.

The STFT of signal $x\left(t\right)$ is defined as
(15)$$STFT_{x}\left(t,f\right)=\int_{-\infty }^{+\infty }x\left(t\right)\phi \left(\tau -t\right)\exp\left(-j2\pi f\tau \right)d\tau ,$$
where $\phi \left(t\right)$ is the window function with a short time width.

The following section will simulate and analyze different neural network inputs and neural network recognition results. The frequency resolution of the STFT is limited by the number of Fourier transform points and the maximum frequency, so the signal was transformed to the baseband or low frequency, and then the STFT was performed to better analyze its time–frequency characteristics. The carrier frequency of the simulated LFM signal was 10 MHz, the bandwidth was 15 MHz, the pulse width was 40 μs, and the pulse repetition period was 100 μs. The JNR of the jamming signal was 10 dB, and the jamming signal was subjected to an STFT with a window length of 256 points and a step distance of 40 points.

The jamming signal simulation parameters are shown in Table 2. 

#### 2.3.1. Only Jamming Signal

When the neural network input has only jamming signals, the input is shown in Figure 4. 

It can be seen from Figure 4 that the five types of jamming shown in Figure 4a,b,h–j can be regarded as jamming signals of the same jamming type, with different parameters. Figure 4d,f can also be regarded as jamming signals of the same jamming type in different bandwidths. The jamming recognition network lacks information such as the bandwidth and carrier frequency of the real signal. Therefore, if only the time–frequency diagram of the jamming signal is used as the input, the neural network will not be able to correctly distinguish the seven types of jamming.

#### 2.3.2. Jamming Plus Echo

In order to introduce real signal information to the network input, the STFT time–frequency diagram of the echo signal was added to the network input. The simulation parameters were the same as when there was only jamming. The inputs are shown in Figure 5. 

The darker parts in Figure 5 are signal echoes, and the brighter parts are jamming signals. From Figure 5, it can be seen that after the echo signal was added, the PDTJ shown in Figure 5d and the WDTJ shown in Figure 5f were clearly distinguished, and the two types of jamming could be separated. However, it can also be seen from the Figure that after the echo was added, the distance false target jamming shown in Figure 5a and the distance gate drag jamming shown in Figure 5h were still inseparable. The speed false target jamming shown in Figure 5b and the speed drag jamming shown in Figure 5i were also inseparable. To distinguish false target-type jamming from towing-type jamming, a comparison of different inter-pulse jamming parameters is needed. Therefore, if only the time–frequency diagram of a single pulse is used as an input, it is not possible to correctly identify whether it is towing jamming or false target jamming.

#### 2.3.3. Multi-Pulse Jamming Plus Echo Fusion

To correctly identify towing jamming, it is necessary to extract the characteristics of multiple consecutive pulses and analyze their change trend. When a CNN is used in the field of traditional image recognition, when the input image of the network is a color picture, the input contains three channels of red, green and blue (RGB). Using the recognition of color pictures for reference, the time–frequency picture of three consecutive pulses of STFT can be used as if they were the three color channels of a unified image to synthesize a picture, which was then input into the network for jamming recognition. The fusion of the time–frequency diagram of the STFT of the three pulse period signals was used as the input. The input is shown in Figure 6. 

It can be seen from Figure 6 that the time–frequency diagrams of jamming signals of different pulse periods are the same for various jamming types, including false target jamming, and the time–frequency diagrams of multiple pulse periods are completely overlapped. The time–frequency diagram of drag-type jamming is not exactly the same as the time–frequency diagram of multiple pulse periods, and it is visualized as a translation combination of a single-period time–frequency diagram. After fusing the time–frequency diagrams of three continuous pulses, false target jamming, and towing jamming also had clear distinctions, and the CNN network was able to be used to identify the jamming.

#### 2.3.4. Multi-Pulse Jamming Plus Original Signal Fusion

One of the most difficult problems in spoofing jamming identification is the identification of jamming in the case of a low JNR. Figure 7 shows the results of distance false target jamming STFT and speed false target jamming STFT when the JNR was −10 dB.

Because the echo signal was usually weaker than the jamming signal, in the case of low JNR, the echo will be submerged by the noise first, and thus only the jamming signal can be seen in the picture. The information reflected in the picture is not enough to determine what kind of jamming this is. The same problem can also be encountered in other cases of deceptive jamming.

The echo signal can provide the real signal’s carrier frequency, bandwidth, time delay, modulation slope, and other information to enable jamming identification. In an intelligent recognition system for radar jamming based on a CNN, the information of the echo signal needs to be extracted from the time–frequency diagram, thereby increasing the system’s JNR requirements. The only difference between the original signal and the echo signal is that it did not contain time delay information. Through the observation and analysis of the STFT image in Figure 6, it can be seen that the intelligent identification method proposed in this paper in which the results of the multi-pulse time–frequency image fusion are used as input does not require the specific time delay information between the true and false signals. Therefore, the original signal can be used to replace the echo signal as part of the jamming identification input. The fusion result of the STFT time–frequency diagram of the original multi-pulse signal plus the STTF time–frequency diagram of the jamming signal can be used as the input of the jamming identification system to correctly identify the jamming type.

As shown in Figure 8 the echo signal could not be detected with a low JNR. By replacing the echo signal with the original signal, the carrier frequency, bandwidth, modulation slope and other information of the real signal could still be obtained, so that it is possible to identify which type of jamming it belongs to.

## 3. Results and Discussion

### 3.1. Generation of Simulation Jamming Signals

In order to make the model have improved generalized performance, this paper simulated and generated a variety of chirp radar signals and generated a variety of simulation signals of 10 kinds of jamming, under different jamming parameters. The chirp signal parameter settings are shown in Table 3. 

The jamming parameter settings are shown in Table 4.

Each of the 10 types of jamming generated 324 jamming signals with different parameters. A total of 35,640 different jamming signals were generated between −13 dB and −3 dB. The STFT of the jamming signal generated a 128 × 244 time–frequency diagram.

Four data sets were generated by the jamming signal, jamming plus echo, multi-pulse jamming plus echo fusion, and multi-pulse jamming plus original signal fusion techniques.

In each data set, 2000 random pictures from 3240 pictures of each type of JNR were used as the test set, 1100 pictures were used as the training set, and 264 pictures were used as the verification set.

### 3.2. Comparison of Recognition Methods

Through the analysis of the jamming signal in Section 2, we can see that when four different time–frequency diagrams were used as input, the types of jamming that could be identified are different. Four different STFT time–frequency diagrams including jamming-only, jamming plus echo, multi-pulse jamming plus echo fusion, and multi-pulse jamming plus original signal fusion, were used as the input of the neural network. We used the same network to test the accuracy of the jamming recognition under different inputs.

Tensorflow is an open source, Python-based machine learning framework. It was developed by Google and has rich applications in graphics classification, audio processing, recommendation systems, and natural language processing. It is the most popular machine learning framework at present. An Nvdia CUDA was used to accelerate GPU computing. CUDA is a parallel computing platform and programming model invented by NVIDIA. It can greatly improve computing performance by using the processing power of the GPU. The initial learning rate of the model was 0.0002, BatchSize = 32, epochs = 20, and dropout = 0.5. Fine-tune transfer learning was used to accelerate convergence. The 22,000-picture input set was divided into the test sets input to the trained network, and the confusion matrix to verify the four methods through simulation was obtained.

According to the confusion matrix of the four methods in Figure 9, when the time–frequency diagram of the jamming signal or the time–frequency diagram of the jamming plus echo signal was used as the input of the network, the network could identify some jamming, but jamming types such as false target jamming and towing jamming could not be identified correctly. When THE multi-pulse jamming plus echo fusion or THE multi-pulse jamming plus original signal fusion time–frequency diagram was used as THE network input, the network could correctly identify all jamming types. The recognition model with ResNet50 network as its core thus had good recognition accuracy.

### 3.3. Improved Method Performance

In order to verify the performance improvement of the method proposed in this paper to replace the echo signal with the original signal in the case of a low JNR, the fusion time–frequency graph data set of the multi-pulse jamming plus echo and the fusion time–frequency graph data set of the multi-pulse jamming plus original signal generated in this paper were respectively input to the network model for training. The jamming to signal ratio (JSR) was 3 dB, and the JNR was −13 dB to −3 dB. Under each different JNR, 200 pictures of each jamming type were input into the network for identification, and the accuracy of the two methods under the different jamming types and different JNRs were obtained. The accuracy curves of the two methods are shown in Figure 10 under different JNRs.

It can be seen from Figure 10 that in the case of high JNR, two methods had higher accuracy rates of jamming recognition. When the JNR decreased, different jamming types interfered, and the accuracy rate of the jamming recognition was reduced. In the case of low JNR, the recognition accuracy of the original signal plus jamming was higher than the recognition accuracy of the echo signal plus jamming.

The transformation curve of the total jamming recognition success rate of the two methods under different JNRs is shown in the figure below.

It can be seen from Figure 11 that both methods achieved excellent results when the JNR was high. Above −4 dB, the recognition accuracy rate reached 99%. But as the JNR drops, the echo signal is drowned out by the noise. The performance of the model using the time–frequency diagram of echo plus jamming as the network input decreased faster, and the performance of the model that used the time–frequency diagram of jamming plus the original signal as input decreased slower. When the multi-pulse original signal plus jamming time–frequency diagram was the input network model the accuracy of jamming recognition was improved under different JNRs. Especially in the case of a low JNR, there was a significant improvement. At −13 dB, the accuracy rate was still 72%. The neural network model of jamming identification has good robustness.

## 4. Conclusions

Based on neural network input research, this paper proposes a method for identifying deceptive jamming of chirp radar. By analyzing the information needed for interference recognition, the input of the neural network was changed. First, the STFT time–frequency diagram of the jamming signal was used as the input of the jamming identification network, which provided the time–frequency information of the jamming signal for the identification network, and achieved the identification of some types of jamming. Then, adding the time–frequency diagram of the echo signal to the network input provided information about the real signal and the difference between the jamming and the real signal for identification by the network, which increased the number of types of jamming that the network recognized. Then, the fusion of multiple-pulse jamming plus echo time–frequency graphs were used as the input of the network, which provided jamming pulse information for identification network and achieved the identification of all interference types. Finally, by replacing the time–frequency diagram of the echo signal with the time–frequency diagram of the original signal as a part of the network input, the accuracy rate of interference recognition increased under low-JNR conditions. By building a model with ResNet50 network as the core, simulation tested the effectiveness of the jamming identification method and the improved method.

## Figures and Tables

**Figure 1 sensors-21-02693-f001:**
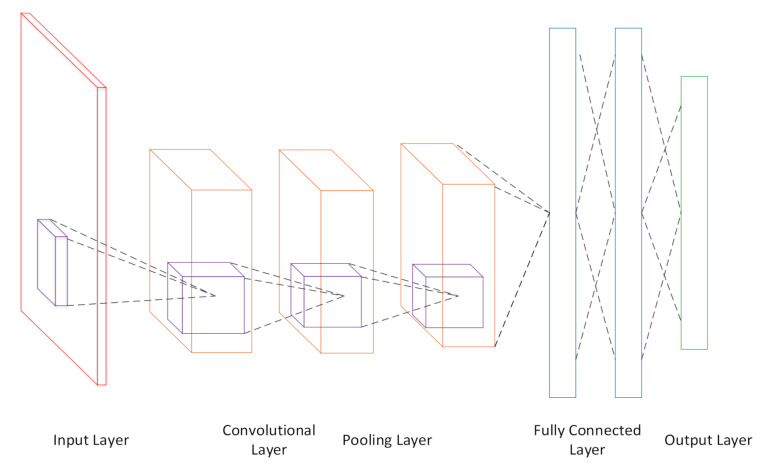
Basic structure of a CNN.

**Figure 2 sensors-21-02693-f002:**
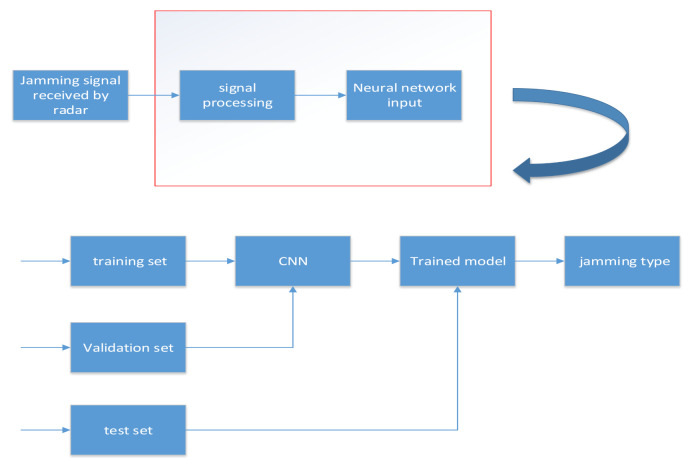
Identification framework.

**Figure 3 sensors-21-02693-f003:**
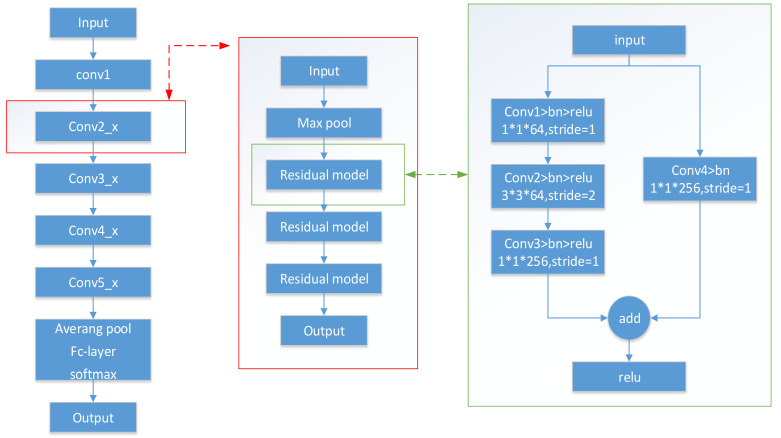
ResNet50 model.

**Figure 4 sensors-21-02693-f004:**
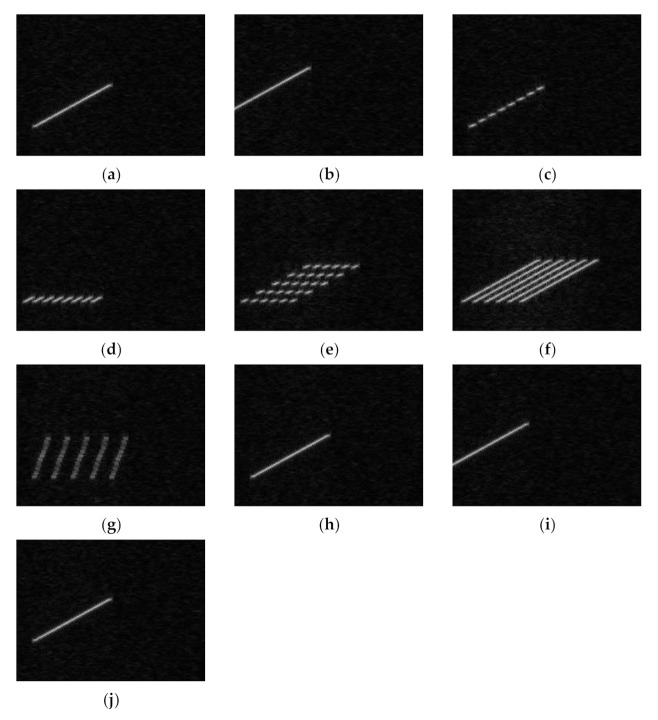
Time–frequency diagram of jamming signals. (**a**) RFTJ. (**b**) VFTJ. (**c**) ISDJ. (**d**) PDTJ. (**e**) ISRJ. (**f**) WDTJ. (**g**) SMSP. (**h**) RGPO. (**i**) VGPO. (**j**) R-VGPO.

**Figure 5 sensors-21-02693-f005:**
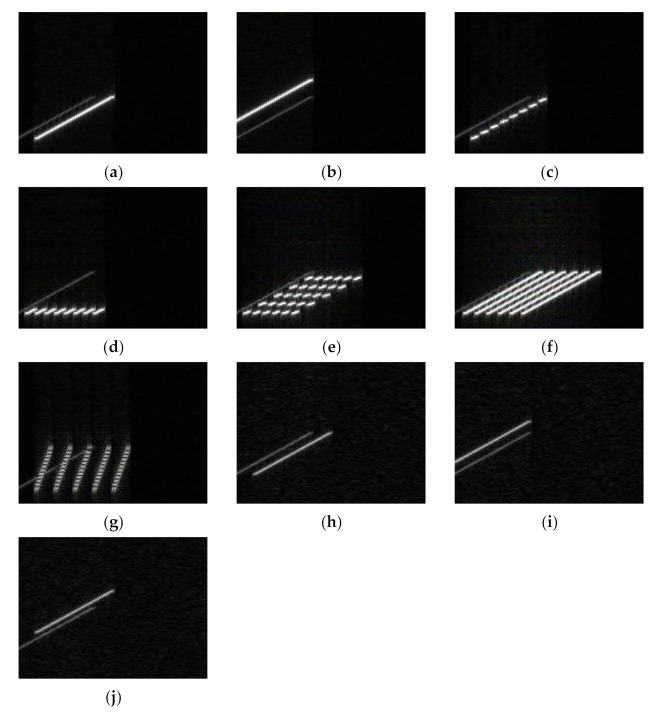
Time–frequency diagram of jamming plus echo. (**a**) RFTJ + echo. (**b**) VFTJ+ echo. (**c**) ISDJ + echo. (**d**) PDTJ + echo. (**e**) ISRJ + echo. (**f**) WDTJ+ echo. (**g**) SMSP + echo. (**h**) RGPO + echo. (**i**) VGPO + echo. (**j**) R-VGPO + echo.

**Figure 6 sensors-21-02693-f006:**
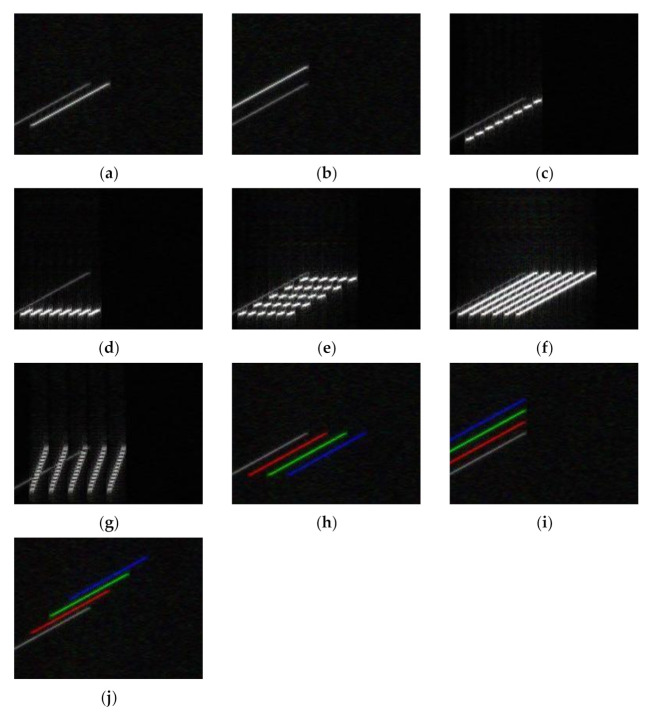
Multi-pulse jamming plus echo time–frequency diagram fusion. (**a**) RFTJ + echo. (**b**) VFTJ + echo. (**c**) ISDJ + echo. (**d**) PDTJ + echo. (**e**) ISRJ + echo. (**f**) WDTJ + echo. (**g**) SMSP + echo. (**h**) RGPO+ echo. (**i**) VGPO + echo. (**j**) R-VGPO+ echo.

**Figure 7 sensors-21-02693-f007:**
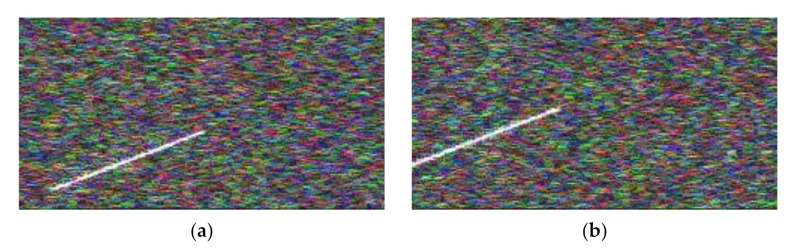
Multi-pulse jamming plus echo time–frequency diagram fusion under low JNR. (**a**) RFTJ + echo. (**b**) VFTJ + echo.

**Figure 8 sensors-21-02693-f008:**
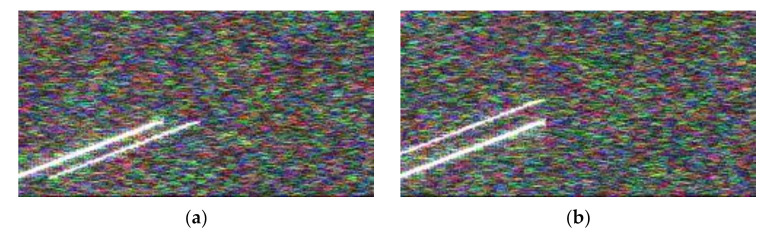
Multi-pulse original signal plus jamming time–frequency diagram fusion under low JNR. (**a**) RFTJ + original signal. (**b**) VFTJ + original signal.

**Figure 9 sensors-21-02693-f009:**
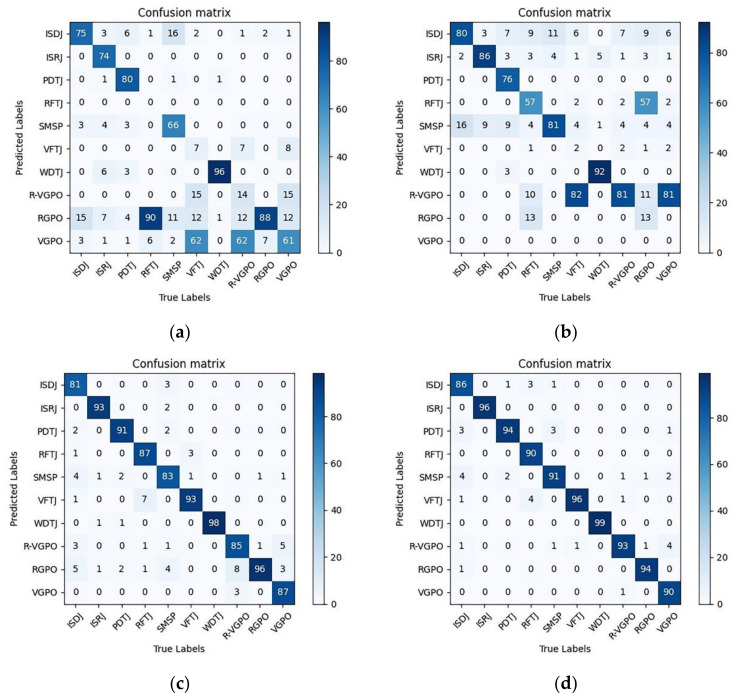
Confusion matrix of the four methods. (**a**) Only jamming signal. (**b**) Jamming plus echo. (**c**) Multi-pulse jamming plus echo fusion. (**d**) Multi-pulse jamming plus original signal fusion.

**Figure 10 sensors-21-02693-f010:**
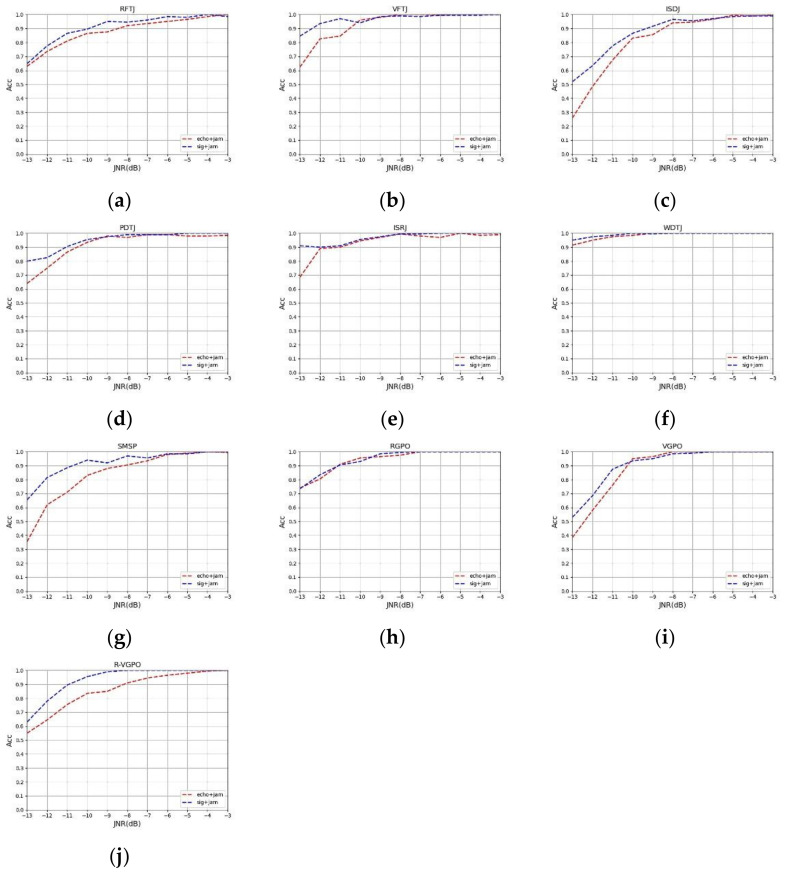
Comparison of recognition accuracy of 10 kinds of jamming between jamming plus echo and jamming plus original signal. (**a**) RFTJ. (**b**) VFTJ. (**c**) ISDJ. (**d**) PDTJ. (**e**) ISRJ. (**f**) WDTJ. (**g**) SMSP. (**h**) RGPO. (**i**) VGPO. (**j**) R-VGPO.

**Figure 11 sensors-21-02693-f011:**
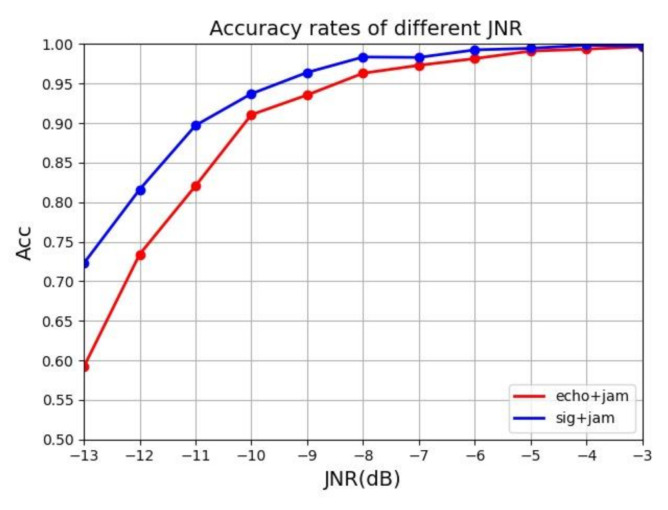
Comparison between jamming plus echo and jamming plus original signal.

**Table 1 sensors-21-02693-t001:** ResNet model parameters.

Layer Name	Output Size	50-Layer
conv1	112 × 112	7 × 7, 64, stride 2
conv2_x	56 × 56	3 × 3 max pool, stride 2
		$\left[\begin{array}{c} 1\times 1.64\\ 3\times 3.64\\ 1\times 1256 \end{array}\right]$×3
conv3_x	28 × 28	$\left[\begin{array}{c} 1\times 1128\\ 3\times 3128\\ 1\times 1512 \end{array}\right]$×4
conv4_x	14 × 14	$\left[\begin{array}{c} 1\times 1256\\ 3\times 3256\\ 1\times 11024 \end{array}\right]$×6
conv5_x	7 × 7	$\left[\begin{array}{c} 1\times 1512\\ 3\times 3512\\ 1\times 12,048 \end{array}\right]$×3
	1 × 1	average pool, 1000-d fc, softmax

**Table 2 sensors-21-02693-t002:** Simulation parameters of jamming signal.

Deceptive Jamming Type	Abbreviated Symbols	Parameter Setting
Range False Target Jamming	RFTJ	Distance between true and false targets: 3000 m
Velocity False Target Jamming	VFTJ	Doppler frequency: 6 MHz
Interrupted-Sampling Directly Jamming	ISDJ	Slice width: 3 μs, Slice period: 5 μs
Partial-pulse Dense Transmit Jamming	PDTJ	Slice width: 5 μs, Number of forwarding: 8
Interrupted-Sampling Repeater Jamming	ISRJ	Slice width: 3 μs, Slice period: 8 μs, Number of forwarding: 6
Whole-Pulse Dense Transmit Jamming	WDTJ	Forwarding cycle: 16 μs Forwarding times: 6
Smeared Spectrum Jamming	SMSP	Frequency modulation slope: 5 times the original FM slope, Forwarding times: 5
Range Gate Pull-off	RGPO	Towing distance: 3000 m/s
Velocity Gate Pull-off	VGPO	Towing speed: 40,000 MHz/s
Range-Velocity Gate Pull-off	R-VGPO	Distance dimension towing speed: 3000 m/s; Velocity dimension towing speed: 60,000 MHz/s

**Table 3 sensors-21-02693-t003:** Chirp signal parameters.

Carrier Frequency (MHz)	Bandwidth (MHz)	Pulse Broadband (μs)	Pulse Period (μs)
0/5/10	5/15/25	20/40/60	100

**Table 4 sensors-21-02693-t004:** Jamming parameters.

Type of Jamming	Parameter Settings
RFTJ	Distance between true and false targets: 3000/2400 m
VFTJ	Doppler frequency4/3 MHz
ISDJ	Slice width: 0.1/0.05 times pw; Slice period: 0.2/0.1 times pw
PDTJ	Slice width: 0.33/0.5 times pw; Number of forwarding: 7/8
ISRJ	Slice width: 0.1/0.05 times pw; Slice period: 0.2/0.1 times pw; Number of forwarding: 6/5
WDTJ	Forwarding interval: 3/6 μs; Number of forwarding: 6/5
SMSP	Frequency modulation slope: 4/5 times the original slope, Number of forwarding: 4/5
RGPO	Towing distance: 3/1.5 km/s
VGPO	Towing speed: 40,000/30,000 MHz/s
R-VGPO	Towing speed: 3/1.5 km/s; Towing speed: 40,000/30,000 MHz/s

## Data Availability

Data is contained within the article. The data presented in this study are available in article.

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
