# Peer review of "Intelligent Recognition of Chirp Radar Deceptive Jamming Based on Multi-Pulse Information Fusion"

_sensors, 2021, doi:10.3390/s21082693_

Round 1

Reviewer 1 Report

VLSI in the first line of introduction should be spelled out!

What do you mean by “In ref [1] and ref [2] ,” look at some similar literature review and follows the same pattern

The same citation follows on, if you think your styleis right please provide some examples

Wrong structure” Based on this, the range gate pull off jamming and the velocity gate pull off jamming are identified.”

“In ref [4], the researchers used the second- and fourth-order cumulants of the radar received
signal as characteristic parameters to identify active jamming” could be “The researchers used the second- and fourth-order cumulants of the radar received
signal as characteristic parameters to identify active jamming [4].”

This is your study contribution “Therefore, this paper studies the input of the neural network.”?

You should work on the study contribution.

Before starting “2. Jamming recognition method” give some introduction what readers are expected to see!

Incomplete sentence “it can be known that when the timefrequency diagram of the jamming signal or the time-frequency diagram of the jamming plus echo signal is used as the input of the network.”

A new subsection in introduction should be added regarding the problem statement and study contributions.

Also make sure there is no incomplete pr grammatically wrong sentence in your next submission. Please get the manuscript proof edited! Best

Reviewer 2 Report

Dear authors,

I appreciate your great efforts involved in this interesting research, very attractive for a large community of readers and specialists working in the field. To increase further the quality and the visibility of this paper, I would like to make  only the following short comments:

  1. To simplify the manuscript reading, please attach a table with the acronyms/abbreviations used inside the text.
  2. The reference section can also be extended with the latest papers in the field (2020, 2021), and also for an increase in the visibility of this journal, it is a good idea to add some papers previously published by the same journal close to the topic of the present research work.
  3. For readers to increase their understanding of the paper content, it is helpful to provide some details about the CNN architecture, ResNet50 (Figure 2), the TensorFlow framework, NVIDIA Cuda used to speed up GPU computation, and also the COCO dataset.
  4. A rigorous robustness analysis for the proposed CNN architecture to the changes in the values ​​of the main structure parameters and noise level in the input data set will be a great challenge.

Thanks, 

Round 2

Reviewer 1 Report

My concerns are addressed